# Impacts of Long COVID on workers: A longitudinal study of employment exit, work hours and mental health in the UK

**Darja Reuschke**[1]*, **Donald Houston**[1], **Paul Sissons**[2]

**1** Birmingham Business School, City-Region Economic Development Institute, University of Birmingham, Birmingham, United Kingdom, **2** Keele Business School, Keele University, Staffordshire, United Kingdom

\* d.reuschke@bham.ac.uk

**Data Availability Statement:** University of Essex, Institute for Social and Economic Research. (2021). Understanding Society: COVID-19 Study, 2020-2021. [data collection]. 11th Edition. UK Data

## Abstract

### Background

The COVID-19 pandemic has had enormous implications for the world of work. However, there has been relatively little focus on the employment and workforce challenges of the virus in relation to workforce health, beyond the immediate management of the spread of the disease. There is an important gap in understanding the ongoing workforce issues created by the significant incidence of Long COVID in the population.

### Aim

This paper examines the effects of Long COVID on employment and workers' mental health to contribute to understanding of work-limiting health conditions and to offer policy implications for COVID-19 and similar health conditions on employment and the workforce.

### Methods

A large national panel study for the UK is used to estimate the likelihood of exiting employment as well as on changes in working hours and general mental health and happiness of those who remain in work. The sample includes individuals 16 years and older who were in employment in January/February 2020 and followed during the pandemic 2020–2021. Long COVID is self-reported in the data. Informed by conceptual consideration of employment protection in the UK, two groups of individuals with Long COVID are defined based on the duration of symptoms. Group 1 has Long COVID 5–28 weeks after an infection with COVID-19, which is up to the maximum length of Statutory Sick Pay in the UK. Group 2 has symptoms for 29+ weeks, which is beyond the statutory entitlement to sickness pay. Panel regression models are fitted both with fixed-effects and random-effects. Individual and job characteristics are used as controls Those with no COVID-19 symptoms are the reference group.

### Results

In between-person comparison, Group 2 is at higher risk of exiting employment compared to those with no COVID-19 symptoms. Between-person estimates of mental health and

Service. SN: 8644, DOI: 10.5255/UKDA-SN-8644-11 University of Essex, Institute for Social and Economic Research. (2022). Understanding Society: Waves 1-11, 2009-2020 and Harmonised BHPS: Waves 1-18, 1991-2009. [data collection]. 15th Edition. UK Data Service. SN: 6614, DOI: 10.5255/UKDA-SN-6614-16.

**Funding:** The author(s) received no specific funding for this work.

**Competing interests:** The authors have declared that no competing interests exist.

well-being show negative effects of Long COVID for both groups but these are greatest in Group 2. Within-person estimates suggest that factors associated with earnings mediate the negative Long COVID effects on mental health in Group 1 and that Group 2 adapts to working with Long COVID. Group 1 is at risk of working zero hours (i.e. being on sick leave) but neither Group 1 nor Group 2 have a higher probability of working fewer hours compared to those with no COVID-19 symptoms. The negative impact of Long COVID on working hours stems primarily from working zero hours (sickness leave) rather than working fewer hours, suggesting a lack of accommodation by employers of Long COVID at work.

## Policy implications

The extension of Statutory Sickness Pay and greater flexibility to manage partial (returns to) work would help preserve employment and mental health. Those with Long COVID for 12 months are likely to meet the definition of disability and so have a right to receive reasonable workplace adjustments.

## 1. Introduction

Long COVID describes a long-term debilitating illness following an infection with the SARS-CoV-2 virus. It is conceptually defined as persistent COVID-19 symptoms or late complications of the virus infection that develop at least five weeks after an initial infection with the virus [1, 2]. Symptoms of Long COVID are multidimensional and can include chronic or episodic physical, cognitive or mental illness [3–5]. Fatigue, breathing problems, inability to concentrate and muscle pain are common [6, 7] and can pose significant limitations on daily activities [8]. Symptoms often fluctuate or relapse [2, 9]. Worldwide an estimated 65 million people have had Long COVID [10]. It affects individuals of all ages [10] but those between 45–54 are at greatest risk [8]. Double vaccination only gives modest protection [11]. It is estimated that 10–12% of the vaccinated population develop Long COVID following a COVID-19 infection [10]. The higher prevalence of Long COVID in people of working-age and the withdrawal of public vaccination programmes for those under 65 years of age in the UK mean that Long COVID is set to have a significant and lasting impact on individuals' health, employment and earnings. There is currently no effective treatment for Long COVID and it can be difficult to clinically diagnose [10].

The onset of long-term illness can have three contrasting, although not mutually exclusive, negative impacts on employment outcomes with consequences for both employees and employers: i) leaving the labour market, either on a permanent or temporary basis until health may recover; ii) reducing hours worked in order to accommodate health problems, including sickness absence; or iii) continuing in existing work arrangements, with possible further consequences for the health and well-being. In addition to the impacts of long-term illness such as Long COVID itself on mental health and job loss, impaired ability to work can have further significant negative impacts on mental health.

Given the large numbers of the population reporting Long COVID [6, 10] and a sharp rise in economic inactivity (those neither working nor seeking or available for work) and resultant labour shortages [12], it is surprising that little robust evidence is available on the potential workforce impacts [1]. Narratives about the impact of the pandemic on the workforce have tended to focus on lifestyle reappraisal such as the 'Great Retirement' and only recently have

begun to acknowledge that the pandemic's direct health impact may be an important driver of rising ill-health and economic inactivity. This paper aims to help address this gap through providing evidence on the impacts of Long COVID on workers using data for the United Kingdom (UK). The focus is on employment outcomes as well as the mental health outcomes of those in employment covering a range of impacts of Long COVID on workers.

The main contribution of this study is to provide robust evidence on the impact of COVID-related health conditions on employment. This is an important contribution to social epidemiology and public health studies, which overwhelmingly focus on the impact of employment on health [13]. The morbidity effects of the COVID-19 pandemic have been documented and, therefore, research is now needed on how developing Long COVID affects employment and in-work experiences. As with other difficult-to-diagnose illnesses with varied symptoms such as chronic fatigue syndrome, research is also needed to reveal and change the societal misperceptions, stigma and discrimination experienced by people and employees with 'unrecognised' diseases where individuals usually do not have a specific diagnosis [14–16]. We use a large national longitudinal survey of people 16 years and older and multi-variate panel regression modelling to estimate how the development of Long COVID has changed individuals' employment and in-work experience.

## 2. Context and existing evidence

Poor health and disability (a health condition that has lasted or is expected to last more than 12 months and limits daily activity) carry large penalties for employment [17]. Chronic illnesses, disabilities and mental health conditions have been associated with labour market exit and barriers for re-employment [18–20]. Disabilities and poor health have also been associated with worse in-work experiences compared to non-disabled employees in relation to contentment, perception of fairness, well-being and job satisfaction [21, 22]. These health-related inequalities in work experiences have been conceptualised through multiple interconnected factors related to individual characteristics (e.g. type of disability), context (legislation) and at the organisational level factors including discrimination and the lack of workplace accommodations and provision of support [23, 24]. Long COVID, as a new complex illness, raises therefore important questions about employment outcomes and in-work experiences. Investigating work outcomes of Long COVID adds to previous studies on work-limiting conditions through focussing on an illness which is often associated, similar to chronic fatigue syndrome, with comorbid health conditions and people often do not receive appropriate treatment and are not eligible for disability benefits [18].

Studies on Long COVID have suggested large negative effects of developing the illness on quality of life due to the major challenges posed by physical health conditions, problems with mental health including anxiety and depression and the unpredictability of symptoms [7, 25–28]. Scientific evidence on the impact of Long COVID on employment and in-work experiences are limited as most existing studies have been based on samples not designed to investigate consequences for work and employment, in particular studies based on samples of hospitalised patients [29] or limited to individuals with COVID-19 symptoms or self-reported Long COVID [7, 9] with no comparator or control group. A limited number of studies have recruited small non-random samples of workers to describe experiences of workers with Long COVID [30, 31]. These scientific limitations notwithstanding, existing evidence suggests that Long COVID creates burdens for the ability to return to work due to the severity of the symptoms and their fluctuating nature [31]. Existing UK studies, based on different sampling strategies, found an increased level of sickness leave due to Long COVID both in hospitalised and not hospitalised patients [7, 32].

Long COVID is associated with stigma as the health conditions are poorly understood and difficult for health services to manage [16]. Many Long COVID sufferers do not have a medical diagnosis [14]. There is no scientific evidence on Long COVID stigma and worker experiences but studies have increasingly shown that health-related stigma by employers, co-workers and others such as health care professionals and customers is a barrier to employment and well-being at work of people with disabilities [33]. A survey by the Trade Union Congress drawing on a self-selective sample of 3,000 respondents with Long COVID, states that 66 per cent of respondents reported they had received unfair treatment at work, and 23 per cent reported their employer had queried whether they had Long COVID or the severity of their symptoms [34].

Long COVID may often limit daily activities for less than 12 months, but it is often long enough to impact on employment and mental health, not least since the UK's Statutory Sick Pay (SSP) system only protects employment for six months (28 weeks) following the onset of a health problem and payment levels are unrelated to normal earnings as in insurance-based schemes. To provide some context, in the UK, only around half of employers have occupational sick pay schemes which pay a higher rate than statutory requirements and few extend beyond six months [35]. SSP is available to employees meeting the health and earnings criteria but who are not covered by an occupational sick pay scheme. SSP is relatively ungenerous (up to £99.35 GBP per week) and is set at a flat rate, rather than providing a level of compensation relative to salary [36]. SSP is also relatively inflexible, creating difficulties around phased returns to work. Longer-term periods of ill-health therefore often precipitate the termination of employment and necessitate a claim to out-of-work sickness benefits [37].

There are also significant differences in access to occupational health support to enable workers to remain in work. Large and public sector employers are more likely to offer access to occupational health services, while lower paid and lower qualified employees, those working in smaller firms, are less likely to have access to such services [38]. In terms of disability support for those in work, there is also evidence to suggest that access to supportive practices are not evenly distributed among the workforce, and that support for invisible disabilities by employers and colleagues may be more limited than for more visible conditions [39].

Together, these gaps in the sick pay system and inequalities in the coverage of occupational health support represent important limitations in the provision of support to allow individuals to remain in work while managing health conditions in the UK. However, no existing study has investigated the association between work outcomes of Long COVID and the length of employment protection in law and how the employment and in-work experiences of Long COVID may be mitigated by certain worker or job characteristics (e.g. skills level, earnings).

## 3. Data and methods

### 3.1 Survey data

The dataset used is the *Understanding Society COVID-19 Study 2020–2021*, which draws on a large household panel study, Understanding Society (USoc), which started in 2009 with a nationally representative sampling frame. The data are available to researchers who are registered with the UK Data Service [40]. The present study has received ethical approval from the Humanities and Social Sciences Committee at the University of Birmingham (ERN_2069).

The first survey round of the USoc COVID-19 Study was conducted in April 2020, soon after the onset of the pandemic, administered online to all adult members 16 years and older of the annual panel study (n = 42,330), and again every month until July 2020, thereafter every two months until March 2021, with a last wave in September 2021 (wave 9). In the first wave, 17,745 individuals completed the survey [41]. Individuals could also join the study in waves

2–4. In total, across all nine waves, 19,763 individuals participated in the study. However, as in all longitudinal surveys, there is an issue with panel attrition. From all individuals who participated in the study, 40.5% participated in every wave. A relatively high proportion of 9.5% only participated in wave 1 (April 2020).

The USoc COVID-19 Study 2020–21 can be linked with the USoc main panel study [42]. This allows the linkage of pre-COVID-19 characteristics to the COVID-19 survey data. We use participant's 2019 interview for additional pre-COVID-19 information. The USoc COVID-19 Study ended before the spike in COVID-19 infections caused by the BA.2 Omicron virus. However, the design and breadth of the USoc COVID-19 Study including information on individuals pre-COVID characteristics, allow an in-depth investigation of employment effects not possible with other secondary survey data.

**3.1.1 Measuring long COVID.**   The study started in April 2020 (wave 1) but only in the three surveys in January, March and September 2021 (waves 7–9) a question on the duration of coronavirus symptoms was included in the questionnaire and asked from those who previously reported that they had COVID-19 symptoms and had not returned to previous health or who reported COVID-19 symptoms in the current survey ("For how many weeks have you experienced coronavirus symptoms?"). In line with the UK Office for National Statistic's Community Infection Survey, we define Long COVID as having had COVID-19 symptoms for five weeks and more [6]. From the initially recruited participants, n = 15,249 were still in the study at least once over the waves 7–9 and n = 9,678 individuals were in the study in each of these three waves our study is based on.

Our focus is on the health-related COVID-19 impact on employment and in-work experiences of those who remain in work. Longer-term periods of ill health that extend the maximum period of Statutory Sickness Pay of 28 weeks in the UK, often precipitate the termination of employment [35]. This is the maximum period we expect especially low-paid workers and those in more insecure employment positions to be more likely to leave employment. Those on lower income and in lower skilled jobs in particular, may need to work even if they are unwell which may result in lower in-work experiences. This is the rationale for deriving two groups of individuals with Long COVID for this study: 1) 5–28 weeks of symptoms and 2) symptoms of 29 weeks and more. For comparison, we define a further two groups of respondents with 3) COVID-19 symptoms of less than five weeks (short COVID) after a COVID-19 infection and 4) a group with no symptoms including recovery from a previous coronavirus infection (used as reference group in the modelling). This is our preferred measure of Long COVID for this study. However, when estimating employment exit, we also use a binary (total) measure of Long COVID (see section 3.3.1). This measure defines as Long COVID all those who reported that they have not recovered from a COVID-19 infection or who had experienced coronavirus symptoms for five weeks and more [32]. The comparison group on this binary measure are respondents with no COVID-19 symptoms or who recovered from a coronavirus infection.

Some people answered in the same interview that they have not recovered from a coronavirus infection which they reported in a previous interview and that they have experienced at the time of the interview coronavirus symptoms for fewer than five weeks. It is likely that these respondents have reported a re-infection with COVID-19 on top of their Long COVID. We do not include these observations in our duration-based Long COVID measure since no reliable information is available on respondent's length of Long COVID.

**3.1.2 Measuring employment and mental health outcomes.**   Respondents were asked about their work status (employed, self-employed, both employed and self-employed, none) in January/February 2020 retrospectively when they first entered the survey and for each wave of the USoc COVID-19 Study. Those who were furloughed were asked to report their work status as being employed. From this we derive a binary variable capturing exit from employment

(paid employment or self-employment) compared to having remained in employment. The survey does not differentiate between economic inactivity and unemployment so that we cannot differentiate between these outcomes.

We further explore employment outcomes for those who remain in employment using number of hours worked to test whether Long COVID is associated with a reduction in working hours.

In addition, two measures of mental health and well-being are examined. From the General Health Questionnaire comprising of 12 questions (GHQ-12) we use the Likert scale ranging from 0 to 36 with 0 indicating the least distressed and 36 the most distressed as a measure of general mental health. From the GHQ-12, the single question on happiness is used as a measure of subjective well-being. It has been used to study performance at work and is therefore useful to examine in-work experiences in our study alongside the full GHQ-12 [43]. The happiness question measures whether individuals feel more happy than usual (coded 1) to much less happy than usual (coded 4). We reverse both the Likert and the happiness scales so that in our modelling findings, positive (negative) values indicate a higher (lower) level of mental health and well-being.

## 3.2 Sample

Our sample draws on survey responses in waves 7–9 (January-September 2021) of the USoc COVID-19 Study in which the question on the duration of COVID-19 symptoms was included. We select respondents who were in employment in January/February 2020. For estimating employment exit, we use a sample of 9,288 individuals with at least one observation between January-September 2021 (Sample 1). For some analysis, we further restrict this sample only to individuals for whom we have information on their employment status in at least two waves (from waves 7–9).

We derive a second sample that includes observations from individuals in employment both in January/February 2020 and in waves 7–9 (Sample 2). The sample size are 20,225 wave-observations (n = 8,708 individuals). The sample size in the multi-variate analysis varies depending on the outcome variable and the modelling frame used.

The second employment sample contains n = 1,906 wave-observations of working zero hours (9.5% of all wave-observations), the average number of number of hours worked of those who worked at least one hour is 34.36 (sd = 13.0), the average reversed GHQ-12 score is 23.6 (sd = 5.8) which is within the range of the expected mental health score, and the highest proportion (48%) of the sample reports that they are no more unhappy or depressed than usual (mean = 3.0, sd = 0.8). A summary of the dependent variables by our duration-based Long COVID variable is shown in Table 1. A sample description of all co-variates used in the multi-variate analysis is shown in the Supplementary Documentation (S1 Appendix).

The sample description in Table 1 shows an increased proportion of respondents not being employed in waves 7–9 among those with Long COVID 29+ weeks. Those with Long COVID symptoms within the Statutory Sickness Pay period (5–28 weeks), have a similar proportion of respondents out of employment compared to those who have no COVID-19 symptoms (including having recovered from a previous COVID-19 infection). In the descriptive data, among respondents who have remained in employment, the proportion of those working zero hours is substantially higher among those with Long COVID 5–28 weeks compared to those with no COVID-19 symptoms although if still working at least one hour per week, this Long COVID group works on average an hour longer than those with no COVID-19 symptoms. Those with Long COVID 29+ weeks have a slightly higher proportion of respondents working zero hours compared to the group of respondents with no COVID-19 symptoms and their

**Table 1. Description of dependent variables by COVID-19 symptoms-duration categories.**

| Dependent variables | No COVID-19 symptoms | Short COVID: <5 weeks | Long COVID: 5–28 weeks | Long COVID: 29 + weeks |
|---|---|---|---|---|
| *Sample 1* | | | | |
| Employment in wave 7–9 (n = 22,106) | | | | |
| Employed | 91.3% | 94.1% | 91.6% | 89.6% |
| Not employed | 8.7% | 5.9% | 8.4% | 10.4% |
| *Sample 2* | | | | |
| Works zero hours in wave 7–9 (n = 19,979) | | | | |
| Yes, works zero hours | 9.2% | 10.8% | 20.2% | 11.1% |
| No, works at least 1 hours | 90.8% | 89.2% | 79.8% | 88.9% |
| Mean number of hours worked in wave 7–9, if worked at least 1 hour (n = 18,073) | 34.3 | 34.7 | 35.3 | 33.2 |
| Mean reversed GHQ-12 score, wave 7–9 (n = 19,659) | 23.79 | 22.64 | 20.15 | 20.82 |
| Mean reversed happiness score, wave 7–9 (n = 19,729) | 3.03 | 2.89 | 2.63 | 2.66 |

Source: Understanding Society COVID-19 Study 2020–2021, waves 7–9, unweighted data

mean working hours (if they are still working) is lowest of all groups. The mean GHQ-12 and happiness scores show large differences across the four groups. For both Long COVID groups, the scores are substantially lower indicating lower mental health and well-being.

## 3.3 Analytical framework

We have panel data (individuals *i* observed at different times *t*) and estimate both random-effects and fixed-effects panel models for each of the set of employment and mental health outcomes. Stata 16 is used for the statistical analysis.

The fixed-effects panel model estimates within-individual variation and requires that the same individual is observed twice, once when they experienced COVID-19 symptoms and once when they did not experience COVID-19 symptoms. These models include an error term that represents individual-specific unmeasured characteristics and therefore allow to control for unobserved heterogeneity due to time-invariant omitted variables and traits. The disadvantage is, however, that individuals in our study who experienced Long COVID in all three subsequent waves, are not included in the statistical analysis.

This is why we also report random-effects models which estimate between-individual variation and within-individual variation. These models, however, assume that the error term is not correlated with unobserved time-invariant (and time-variant) variables. Random-effects models allow the investigation of time-invariant characteristics which we also report. We include as time-invariant variables pre-COVID-19 job characteristics of the respondents which we link to the survey data from the main panel study.

**3.3.1 Employment exit.** We estimate the probability of having moved from employment to non-employment with logistic regression models. Unfortunately, we cannot fit a logistic regression model with fixed effects using our duration-based measure. The model requires that people have experienced between subsequent waves changes in the work status and COVID-19 symptoms categories. We can fit a logistic regression model with fixed effects using the binary (total) Long COVID measure (COVID-19 symptoms 5+ weeks vs <5 weeks including no symptoms and recovery from COVID-19). We report random-effects findings for the duration-based measure of Long COVID (no symptom, short COVID, Long COVID 5–28 weeks, Long COVID 29+ weeks).

**3.3.2 Working time.** We estimate effects of Long COVID on working hours using a two-step approach. First, we use a binary logistic regression with working zero hours as dependent variable and, second, a log-linear regression model of working hours.

**3.3.3 Mental health and well-being.** We regress the reversed and standardised GHQ-12 score using linear panel regressions (positive/negative values indicate better/lower mental health, see 3.1.2). For the reversed 4-item happiness scale which measures whether respondents are less happy or much happier compared to their usual status, ordered logistic panel regression models are fitted.

**3.3.4 Co-variates.** In the fixed-effects regression panel models, co-variates included are the age of the respondents and age squared, two variables indicating whether a partner/spouse and a child 0–15 years of age live in the household, and wave dummies. Time-invariant characteristics are included in the error term.

For estimating the effects of Long COVID on the number of hours worked and mental health and well-being, the following (time-variant) variables are also added in the fixed-effects models: industry sectors and employment status (employee vs self-employed), based on the large variations of working conditions across sectors and the disproportionate impact of the pandemic on self-employed work (note that these are not added in the model of working zero hours due to the small number of observations). In models of mental health and well-being, we further include as a co-variate hourly earnings as lower incomes have been associated with poorer mental health [44].

In the random-effects panel regression models, a comprehensive set of co-variates of personal characteristics is added to the above time-variant variables: sex, highest education, ethnicity, region, and a long-standing health condition in the year preceding the pandemic (from the main panel interview) (see S1 Appendix). Previous research has shown that the prevalence of Long COVID is greater in people with another activity-limiting health condition or disability, in particular asthma, lung disease or heart disease [45].

In the employment exit random-effects models, we further add as pre-COVID-19 characteristics: industry sector, number of hours worked, a temporary (versus permanent) contract, self-employment and hourly earnings, taken either from the 2019 main panel interview or from retrospective information for January/February 2020 collected in the USoc COVID-19 survey.

**3.3.5 Robustness checks.** As robustness checks for models of hours worked and mental health, we derive measures of change between the respondent's interview that fell into waves 7–9 and their number of hours work in January/February 2020 and their GHQ-12 score in 2019 (from the USoc main panel interview). The GHQ-12 change score is standardised and reversed so that negative (positive) values indicate a decline (increase) in mental health. Linear regression models are used to examine whether Long COVID is related with a reduction in hours worked or decline in mental health for the same individuals including controls (3.3.4).

## 4. Results

### 4.1 Employment exit

Findings on whether Long COVID is connected with an increased risk of leaving the workforce, are summarised in two models in Table 2. The data, unfortunately, do not allow the investigation of Long COVID using the duration-based measure of Long COVID in a fixed-effects (FE) model framework (which requires change in work status and in the four COVID-19 symptoms groups). However, a fixed-effects logistic regression model is fitted using the binary measure of Long COVID (COVID-19 symptoms 5+ week or not). These results are displayed in Model 1 (Table 2). Model 2 (Table 2) uses the duration-based measure of Long

**Table 2. Employment exit, odds ratios and standard errors.**

| Co-variates | M1 –FE | | M2 –RE | |
|---|---|---|---|---|
| Long COVID 5+ weeks | 1.360 | 0.707 | - | |
| COVID-19 symptoms (Ref. none) | | | | |
| <5 weeks | - | | 1.209 | 0.297 |
| 5–28 weeks | - | | 1.381 | 0.707 |
| 29+ weeks | - | | 3.523* | 1.959 |
| Age | 0.190*** | 0.082 | 0.627*** | 0.209 |
| Age$^2$ | 1.018*** | 0.004 | 1.006*** | 0.001 |
| Child 0–15 | 3.431* | 2.129 | 0.492** | 0.122 |
| No partner (Ref. couple) | 0.929 | 0.370 | 1.169 | 0.222 |
| Female | - | | 0.751 | 0.155 |
| Ethnicity (Ref. White British)[1] | | | | |
| White, other | - | | 1.824 | 0.766 |
| Asian | - | | 3.182** | 1.192 |
| Black | - | | 2.381 | 1.301 |
| Qualification (Ref. high) | | | | |
| Medium | - | | 1.044 | 0.209 |
| Low | - | | 1.119 | 0.394 |
| *Pre-pandemic characteristics (2019/2020)* | | | | |
| Long-standing health cond. | - | | 1.717** | 0.326 |
| Temporary job | - | | 2.712*** | 0.742 |
| Self-employed | - | | 0.170*** | 0.059 |
| Industry (Ref. SIC-KLMN)[1,2] | | | | |
| SIC CDEF | - | | 1.435 | 0.532 |
| SIC GI | - | | 1.529 | 0.524 |
| SIC HJ | - | | 2.870** | 1.112 |
| SIC OPQ | - | | 1.257 | 0.361 |
| SIC RSTU | - | | 1.022 | 0.484 |
| Hours worked | - | | 0.926*** | 0.008 |
| Net earnings/hour (logged) | - | | 0.764 | 0.128 |
| Wave-obs. | 1,142 | | 16,735 | |
| Individuals | 407 | | 7,085 | |
| LR Chi$^2$(df) / Wald Chi$^2$(df) | 58.9(7) | | 408.8(39) | |

Source: USoc COVID-19 Study 2020–2021, waves 7–9. Sample of respondents who were in work in Jan/Feb 2020. Wave dummies are included in all models and region dummies in the RE models.

Significance level

***p<0.001

**p<0.01

*p<0.05.

[1] Categories with small cell sizes not shown.

[2] CDEF-manufacturing, electricity, gas, water, construction; GI-retail, wholesale, hospitality; HJ-transport, information and communication; KLMN-finance, insurance, professional & admin. Services; OPQ-public admin., health, social, education; RSTU-other personal services.

COVID with no COVID-19 symptoms as reference group in the random-effects panel framework. Results are shown in odds ratio (OR). The number of wave-observations and of individuals between the fixed-effects (FE) and random-effects (RE) model are strikingly different as the fixed-effects models only uses wave-observations from individuals who experienced both a change in work status and Long COVID. The random-effects model instead estimates the

odds of an employment exit for the four COVID-19 symptoms groups and controls. Odds ratios larger (smaller) than 1 indicate that an employment exit has greater (fewer) odds of occurring with the co-variate.

The binary (total) measure of Long COVID is not significantly related with an exit from employment in the fixed-effects model (Model 1, Table 2). The odds ratio is increased (OR = 1.36, sd = 0.7) but not significantly larger than Zero. The duration-based measure of COVID-19 symptoms instead shows a large and significant increase of the odds of those with Long COVID 29+ weeks (OR = 3.5, sd = 1.9) to have exited the workforce compared to those who did not have COVID-19 symptoms or have recovered from a COVID-19 infection (Model 2, Table 2). In comparison, those with Long COVID 5–28 weeks (that is within the Statutory Sickness Pay period) and those with short COVID (<5 weeks) do not have significantly higher odds of not being in employment any more than those with no COVID-19 symptoms.

Long COVID and a previous long-standing health condition are interrelated in our sample (e.g. 53% of respondents with COVID-19 symptoms for 29+ weeks reported a previous health condition compared to 28% of respondents with no COVID-19 symptoms). Previous research has also shown that the prevalence of Long COVID is greater in people with another activity-limiting health condition or disability, in particular asthma, lung disease or heart disease [45]. Importantly, however, findings in Model 2 in Table 2 show that the odds of an employment exit are twice as high for Long COVID for 29+weeks to a previous long-standing health condition (OR = 3.5, sd = 1.9 compared to OR = 1.7, sd = 0.3).

In both model frameworks, findings show an inverse relationship with age suggesting that both young and older people left the workforce. Furthermore, the random-effects model framework (Model 2) reveals an increased risk of employment exit of Asian people (OR = 3.2, sd = 1.2), those with insecure (temporary) jobs (OR = 2.7, sd = 0.7) or those who worked fewer hours before the outbreak of the pandemic (OR = 0.9, sd = 0.01). The only industry sector that shows a significant positive relationship with an employment exit compared to the advanced service sector is the combined category of transport, information and communication (OR = 2.9, sd = 1.1).

## 4.2 Working hours

Findings in Table 3 show whether Long COVID has had a negative impact on the number of hours worked (of those still in employment). Models 1 and 2 estimate the risk of working zero hours using fixed-effects (FE) and random-effects (RE). Results are shown in odds ratio. Using observations from individuals who worked at least one hour in the reference week (see Table 1), Models 3 and 4 estimate the effect of Long COVID and controls on number of working hours (logged) first with fixed-effects (FE) and then with random-effects (RE).

We find that having short COVID (symptoms <5 weeks) or Long COVID 5–28 weeks, i.e. a duration covered by Statutory Sickness Pay according to employment protection law, increases the odds of not working (while still being in employment) (OR = 1.9–3.6, sd = 0.3–1.0). For those with Long COVID beyond the maximum period of employment protection in the UK of 28 weeks, we do not find a relationship with working zero hours.

In the random-effects model (Model 2, Table 3) also a previous long-standing health condition shows a positive relationship with working zero hours (OR = 1.7, sd = 0.2), although this effect is smaller relative to having short COVID (OR = 2.1, sd = 0.3) and Long COVID 5–28 weeks (OR = 3.6, sd = 1.0). The random-effects model also reveals a greater risk of working zero hours in industry sectors particularly affected by lockdowns (hospitality—GI, transportation—HJ and personal services—RSTU), women (OR = 1.9, sd = 0.2), the self-employed (OR = 2.96, sd = 0.5) and those on lower earnings (OR = 0.76, sd = 0.8).

**Table 3. Zero hours worked (odds ratios) and working hours (coefficients).**

| Co-variates | Works zero hours–OR | | Working hours (logged)–Coeff. | |
|---|---|---|---|---|
| | M1 –FE | M2 –RE | M3 –FE | M4 –RE |
| COVID-19 symptoms (Ref. none) | | | | |
| <5 weeks | 1.898*** (0.331) | 2.137*** (0.318) | -0.012 (0.011) | 0.016 (0.010) |
| 5–28 weeks | 2.145* (0.701) | 3.614*** (1.042) | -0.033 (0.027) | -0.017 (0.024) |
| 29+ weeks | 0.704 (0.406) | 1.129 (0.508) | 0.018 (0.039) | 0.011 (0.033) |
| Age | 0.840 (0.307) | 0.766*** (0.022) | 0.072** (0.022) | 0.020*** (0.003) |
| Age$^2$ | 1.003 (0.004) | 1.003*** (0.001) | -0.001*** (0.0002) | -0.0003*** (0.0001) |
| Child 0–15 | 4.442*** (1.473) | 1.510** (0.193) | 0.00003 (0.021) | -0.011 (0.010) |
| No partner (Ref. couple) | 0.884 (0.269) | 0.744* (0.096) | 0.0002 (0.018) | 0.022* (0.010) |
| Industry (Ref. SIC-KLMN)[2] | | | | |
| SIC AB | - | 1.278 (0.659) | -0.046 (0.202) | 0.054 (0.038) |
| SIC CDEF | - | 1.062 (0.257) | 0.016 (0.084) | 0.003 (0.016) |
| SIC GI | - | 8.749*** (1.858) | -0.199** (0.074) | -0.090*** (0.017) |
| SIC HJ | - | 2.946*** (0.721) | 0.006 (0.086) | -0.004) (0.018) |
| SIC OPQ | - | 1.245 (0.227) | 0.030 (0.067) | -0.007 (0.013) |
| SIC RSTU | - | 3.672*** (0.774) | 0.138 (0.080) | -0.080*** (0.016) |
| Self-employed | - | 2.962*** (0.480) | -0.191*** (0.045) | -0.270*** (0.014) |
| Female | - | 1.889*** (0.235) | - | -0.030** (0.010) |
| Ethnicity (Ref. White British)[1] | | | | |
| White, other | - | 0.967 (0.268) | - | -0.024 (0.021) |
| Asian | - | 0.723 (0.193) | - | 0.012 (0.019) |
| Black | - | 0.823 (0.341) | - | 0.032 (0.030) |
| Qualification (Ref. high) | | | | |
| Medium | - | 1.319* (0.161) | - | 0.003 (0.010) |
| Low | - | 1.044 (0.250) | - | 0.020 (0.020) |
| *Pre-pandemic characteristics (2019/2020)* | | | | |
| Long-standing health cond. | - | 1.672*** (0.204) | - | -0.015 (0.010) |
| Net earnings/hour (logged) | - | 0.763** (0.765) | - | 0.047*** (0.009) |
| Hours worked | - | - | - | 0.028*** (0.0004) |
| Wave-obs. | 2,644 | 16,622 | 17,956 | 15,098 |
| Individuals | 954 | 7,213 | 8,182 | 6,922 |
| LR Chi2 / Wald Chi2 / F | 214.5(9) | 482.7(37) | 9.04(8) | 7421.8(38) |

Source: USoc COVID-19 Study 2020–2021. Sample of respondents who were in work in Jan/Feb 2020 and are still in work at t (waves 7–9). Models include wave dummies and the RE models region dummies. Standard errors in brackets.

Significance level

***p<0.001

**p<0.01

*p<0.05.

[1]Categories with small cell sizes not shown.

[2]AB- agriculture, mining; CDEF-manufacturing, electricity, gas, water, construction; GI-retail, wholesale, hospitality; HJ-transport, information and communication; KLMN-finance, insurance, professional & admin. Services; OPQ-public admin., health, social, education; RSTU-other personal services.

Considering only those who are still working at least one hour per week (Models 3 and 4 in Table 3), neither of the COVID-19 symptoms groups display a relationship with the number of hours worked compared to those with no COVID-19 symptoms. Hence, the impact of Long COVID on working hours is due to workers not working at all rather than working reduced hours. This result is supported by additional analysis of the change in working hours between

the COVID-19 surveys (waves 7–9) and January/February 2020 using the duration-based Long COVID variable and the same controls (see S1 Appendix). This shows that those with short COVID (<5 weeks) and Long COVID 5–28 weeks are associated with a significant reduction in working hours compared to those with no COVID-19 symptoms but those with Long COVID 29+ weeks do not work less (or more) than they did before the pandemic.

Furthermore, a previous long-standing health condition is also not related with reduced working hours (Model 4, Table 3). Instead we find that young people (age squared = -0.0003, sd = 0.0001), women (-0.03, sd = 0.01), working in retail and hospitality (GI—-0.09, sd = 0.02) and being self-employed (-0.27, sd = 0.01) are related with shorter working hours.

## 4.3 Mental health and well-being

Table 4 presents findings on the impact of Long COVID on worker's mental health and well-being using the standardised GHQ-12 Likert score (Models 1–3) and the happiness scale from the GHQ-12 (Models 4–6). Two fixed-effects (FE) models are displayed for each outcome variable (Models 1 and 2, 4 and 5) whereby the second model for each adds hourly net earnings to the first model to control for the impact of lower (higher) income on mental health and well-being. Coefficients are displayed for the GHQ-12 score and odds ratios for the happiness scale.

We first consider findings for the GHQ-12. The first fixed-effects model shows a large negative and significant effect of Long COVID 5–28 weeks (-0.23, sd = 0.06) on mental health. This effect is not significant once earnings are controlled for (Model 2, Table 4). Suffering Long COVID for 29+ weeks is not significant in either fixed-effects model. Short COVID (<5 weeks) is significantly negatively associated with mental health in the fixed-effects models (-0.08, sd = 0.02). The short COVID effect on mental health is not influenced by earnings and remains statistically significant in Model 2 (-0.08, sd = 0.03).

The random-effects model (Model 3) (which assumes that the co-variates are not correlated with omitted variables) instead shows for all three COVID-19 symptom groups a large negative and significant relationship with mental health when earnings are controlled for (-0.13, sd = 0.03 - -0.35, sd = 0.08)–and this negative relationship is largest for the Long COVID group with a longer duration of the illness (29+ weeks). For them, the mental health score is reduced by 35% (-0.35, sd = 0.08)–and this effect is large when compared to all other co-variates.

Additional analysis in Table 5 confirms these results and aids interpretation. The dependent variable in Table 5 measures the change of individual's GHQ-12 score between 2019 and during the pandemic surveys (in waves 7–9). The GHQ-12 for the year 2019 is linked to the COVID-19 surveys from the Understanding Society main panel interview. A negative (positive) value indicates a reduction (increase) in mental health. Otherwise the same models including all co-variates are used as in Table 4. Results from random-effects models (Model 4, Table 5) show that those with Long COVID 29+ weeks are also at risk of experiencing the greatest reduction in mental health after earnings are controlled for (-0.195, sd = 0.09) compared to not having COVID-19 symptoms. The effect is smaller for short COVID (-0.13, sd = 0.03) or Long COVID 5–28 weeks (-0.09, sd = 0.06). In the fixed-effects models (Models 1 and 2, Table 5), Long COVID 29+ weeks does not show an effect on mental health. A possible explanation of the difference between the random-effects and fixed-effects estimates may be that people adapt to working with the illness.

When the change in the GHQ-12 score between 2019 and during the pandemic is estimated, the Long COVID group with symptoms 5–28 weeks has significantly poorer mental health before earnings are controlled for (-0.25, sd = 0.05–0.06) (Models 1 and 3, Table 5) but there is no significant Long COVID effect after earnings are included in the models (Models 2

**Table 4. Standardised GHQ-12 score (coefficients) and happiness (odds ratios).**

| Co-variates | GHQ-12 –coeff. | | | Happiness–odds ratio[1] | | |
|---|---|---|---|---|---|---|
| | **M1 –FE** | **M2 –FE** | **M3 –RE** | **M4 –FE** | **M5 –FE** | **M6 –RE** |
| COVID-19 symptoms (Ref. none) | | | | | | |
| <5 weeks | -0.080** (0.024) | -0.079** (0.028) | -0.128*** (0.025) | 0.862 (0.081) | 0.840 (0.092) | 0.731*** (0.064) |
| 5–28 weeks | -0.230*** (0.056) | -0.079 (0.070) | -0.266*** (0.059) | 0.496** (0.107) | 0.659 (0.167) | 0.445*** (0.089) |
| 29+ weeks | -0.089 (0.084) | -0.089 (0.098) | -0.354*** (0.079) | 0.509* (0.161) | 0.620 (0.214) | 0.295*** (0.079) |
| Age | 0.007 (0.049) | 0.036 (0.058) | -0.005 (0.059) | 0.956 (0.208) | 0.934 (0.254) | 0.995 (0.019) |
| Age[2] | 0.0001 (0.0005) | -0.0003 (0.0006) | 0.0001* (0.0001) | 1.005* (0.002) | 1.005 (0.003) | 1.0003 (0.0002) |
| Child 0–15 | 0.002 (0.044) | -0.001 (0.055) | -0.065** (0.022) | 0.874 (0.152) | 0.864 (0.188) | 0.824* (0.062) |
| No partner (Ref. couple) | -0.016 (0.040) | -0.044 (0.048) | -0.175*** (0.023) | 0.924 (0.146) | 0.966 (0.180) | 0.554*** (0.042) |
| Industry (Ref. SIC-KLMN)[2,3] | | | | | | |
| SIC CDEF | 0.013 (0.185) | -0.211 (0.232) | 0.001 (0.039) | 1.783 (1.284) | 0.725 (0.667) | 1.175 (0.149) |
| SIC GI | -0.011 (0.163) | -0.189 (0.204) | -0.057 (0.040) | 0.827 (0.542) | 0.696 (0.520) | 0.967 (0.126) |
| SIC HJ | -0.087 (0.193) | -0.283 (0.226) | -0.074 (0.043) | 0.982 (0.740) | 0.485 (0.416) | 0.899 (0.128) |
| SIC OPQ | -0.188 (0.152) | -0.459* (0.188) | -0.032 (0.029) | 0.343 (0.225) | 0.117** (0.093) | 1.004 (0.098) |
| SIC RSTU | 0.127 (0.187) | -0.101 (0.229) | -0.006 (0.039) | 1.617 (1.104) | 0.842 (0.737) | 0.991 (0.125) |
| Self-employed | 0.004 (0.101) | -0.094 (0.125) | 0.041 (0.034) | 1.238 (0.539) | 1.103 (0.604) | 1.138 (0.128) |
| Net earnings/hour (logged) | - | 0.038* (0.018) | 0.063*** (0.013) | - | 1.060 (0.080) | 1.156** (0.054) |
| Female | - | - | -0.178*** (0.022) | - | - | 0.610*** (0.045) |
| Ethnicity (Ref. White British)[2] | | | | | | |
| White, other | - | - | -0.007 (0.051) | - | - | 0.988 (0.163) |
| Asian | - | - | -0.00002 (0.045) | - | - | 0.978 (0.144) |
| Black | - | - | 0.174* (0.071) | - | - | 2.068** (0.485) |
| Qualification (Ref. high) | | | | | | |
| Medium | - | - | 0.052* (0.023) | - | - | 1.110 (0.082) |
| Low | - | - | 0.130*** (0.048) | - | - | 1.319 (0.208) |
| Long-standing health cond. (2019) | - | - | -0.262*** (0.023) | - | - | 0.434*** (0.033) |
| Wave-obs. | 19,514 | 15,042 | 13,841 | 11,562 | 8,168 | 13,864 |
| Individuals | 8,422 | 7,094 | 6,591 | 3,422 | 2,493 | 6,595 |
| F / Wald Chi2(df) | 13.72 | 8.90 | 658.4(37) | 95.23 (13) | 66.27 (14) | 534.8(37) |

Source: USoc COVID-19 Study 2020–2021. Sample of respondents who were in work in Jan/Feb 2020 and are still in work at t (waves 7–9). The original scores are reversed. Positive (negative) coefficients and ORs >1 (<1) indicate better (poorer) mental health and well-being. Wave dummies are included in all models and region dummies in the RE models. Standard errors in brackets.

Significance level

***$p < 0.001$

**$p < 0.01$

*$p < 0.05$.

[1] Cut-off points not shown.

[2] Categories with small cell sizes not shown.

[3] CDEF-manufacturing, electricity, gas, water, construction; GI-retail, wholesale, hospitality; HJ-transport, information and communication; KLMN-finance, insurance, professional & admin. Services; OPQ-public admin., health, social, education; RSTU-other personal services.

and 4, Table 5). Together with findings for this group in Table 4, earnings are likely to capture that in some (higher paid) jobs tasks are less physical and can be performed better with Long COVID symptoms or that employers may be more likely or better able to accommodate the needs of Long COVID sufferers [31].

Considering now the happiness findings in Table 4 (Models 4–6), these broadly confirm findings of the GHQ-12 mental health score. In the fixed-effects model, both Long COVID

**Table 5. Change in GHQ-12 between COVID-19 surveys (Jan-Sep 2021) and 2019, fixed-effects (FE) and random-effects (RE) panel models, coefficients and standard errors[1].**

| Co-variates | FE Model | | RE Model | |
|---|---|---|---|---|
| | M1 | M2 | M3 | M4 |
| COVID-19 symptoms (Ref. none) | | | | |
| <5 weeks | -0.092*** (0.026) | -0.097** (0.030) | -0.117*** (0.023) | -0.127*** (0.027) |
| 5–28 weeks | -0.248*** (0.061) | -0.124 (0.075) | -0.254*** (0.054) | -0.085 (0.064) |
| 29+ weeks | -0.100 (0.091) | -0.124 (0.107) | -0.193* (0.077) | -0.195* (0.086) |
| Age | -0.012 (0.053) | 0.022 (0.063) | 0.003 (0.006) | 0.005 (0.007) |
| Age$^2$ | 0.0003 (0.0005) | -0.0001 (0.0006) | -0.00002 (0.00006) | -0.00006 (0.00007) |
| Child 0–15 (Ref. no) | 0.017 (0.047) | 0.017 (0.059) | -0.031 (0.023) | -0.059* (0.025) |
| No partner (Ref. couple) | 0.013 (0.044) | -0.016 (0.052) | -0.018 (0.023) | -0.043 (0.025) |
| Industry (Ref. SIC-KLMN)[2,3] | | | | |
| SIC CDEF | 0.092 (0.201) | -0.180 (0.247) | -0.072 (0.040) | -0.085 (0.044) |
| SIC GI | 0.030 (0.175) | -0.167 (0.215) | -0.093* (0.040) | -0.070 (0.045) |
| SIC HJ | 0.009 (0.210) | -0.259 (0.240) | -0.046 (0.045) | -0.042 (0.049) |
| SIC OPQ | -0.125 (0.165) | -0.431* (0.203) | -0.028 (0.031) | -0.030 (0.034) |
| SIC RSTU | 0.183 (0.202) | -0.063 (0.242) | -0.054 (0.039) | -0.006 (0.043) |
| Self-employed (yes) | -0.009 (0.110) | -0.061 (0.134) | -0.065* (0.033) | 0.020 (0.038) |
| Net earnings/hour (logged) | - | 0.041* (0.019) | - | 0.031* (0.015) |
| Female (Ref. male) | - | - | -0.060** (0.023) | -0.045 (0.025) |
| Ethnicity (Ref. White British)[2] | | | | |
| White, other | - | - | 0.027 (0.052) | 0.020 (0.058) |
| Asian or Asian British | - | - | 0.081 (0.047) | 0.099 (0.052) |
| Black or Black British | - | - | 0.060 (0.074) | 0.005 (0.081) |
| Highest qualification (Ref. high) | | | | |
| Medium | - | - | 0.022 (0.023) | 0.053* (0.026) |
| Low | - | - | 0.089 (0.048) | 0.119* (0.055) |
| Long-standing health cond. (2019) | - | - | 0.073** (0.024) | 0.105*** (0.027) |
| Constant | -0.226 (1.407) | -0.673 (1.649) | -0.062 (0.147) | -0.129 (0.170) |
| Wave-obs. | 18,502 | 14,273 | 17,752 | 13,712 |
| Individuals | 7,896 | 6,664 | 7,726 | 6,514 |
| F/Wald Chi2 | 12.94 | 8.43 | 258.98(36) | 224.84(37) |

Source: Understanding Society COVID-19 Study 2020–2021. Sample of respondents who were in work in Jan/Feb 2020 and are still in work at t (waves 7–9). Models include wave dummies and the RE models also region dummies.

Significance level

***p<0.001

**p<0.01

*p<0.05

[1]The change score is reversed and standardised. Positive (negative) values indicate an increase (decrease) in mental health.

[2]Categories with small cell sizes not shown.

[3]CDEF-manufacturing, electricity, gas, water, construction; GI-retail, wholesale, hospitality; HJ-transport, information and communication; KLMN-finance, insurance, professional & admin. Services; OPQ-public admin., health, social, education; RSTU-other personal services.

groups are affected by lower happiness before earnings are controlled for (OR = 0.50, sd = 0.1) (Model 4). The Long COVID effect is reduced and is not statistically significant anymore when earnings are controlled for (Model 5). In the random-effects model (Model 6), compared to the group with no COVID-19 symptoms, short COVID and Long COVID 5–28 weeks are associated with lower happiness (OR = 0.7, sd = 0.06 and OR = 0.44, sd = 0.09) while the odds

ratio is smallest, indicating the greatest reduction in mental health, in the group with Long COVID of 29+ weeks (OR = 0.3, sd = 0.08).

As concerns other co-variates in Table 4, the fixed-effects models show negative effects on mental health and well-being of working in administration, human health and education (OPQ: -0.46, sd = 0.2; OR = 0.12, sd = 0.09) which is likely to reflect the increased job demand in these sectors. The random-effects models further show lower mental health and well-being of women compared to men (-0.18, sd = 0.02; OR = 0.61, sd = 0.05), a finding reported for the COVID-19 pandemic in existing studies [46].

## 5. Conclusions

Our findings provide evidence in between-group comparison of an impact of Long COVID on employment exit for those who have suffered from COVID-19 symptoms beyond the maximum period of statutory sickness leave (which is 28 weeks in the UK).

For those who remain in employment, we observe a negative impact of Long COVID on working hours. This stems primarily from working zero hours (sickness leave) and not from working fewer hours. As previously shown, working fewer hours can help accommodate a work-limiting disability [47] and therefore this finding may suggest a lack of accommodation by employers of Long COVID at work.

The in-work experiences of those with Long COVID were further investigated with the GHQ-12. We find large negative effects in between-group comparison on worker's general mental health and their happiness for the Long COVID group with symptoms for 29+ weeks (beyond the maximum period of employment protection in the case of sickness leave). For this group, analysis of variation within the same individuals shows no negative Long COVID effect on mental health and well-being. We suggest that adaptations of how to live and work with Long COVID could be one possible explanation. Findings for the Long COVID group with symptoms 5–28 weeks are more mixed but suggest that negative impact on mental health is mediated by earnings and the possible job characteristics associated with higher/lower incomes. The findings therefore raise concerns about how workers on lower incomes cope with Long COVID in the workplace.

Despite the strength of our data (large population sample allowing for comparison group analysis, longitudinal design, information on pre-pandemic characteristics of the respondents), there are also limitations. The measure of weeks of COVID-19 symptoms since infection meant that we could derive a measure of Long COVID that incorporates risk of loss of employment. However, there may be inconsistencies in how people who fall 'between' full weeks have answered the question and therefore some people may be misclassified in this study. Some respondents with Long COVID who got reinfected with COVID-19 also did not answer the question on the duration of COVID-19 symptoms for their first infection. Given the study design, we were only able to investigate short-term effects of developing Long COVID on employment and mental health but not the longer-term effects of Long COVID on people's likelihood to return to, or remain in, work. Since Long COVID could be identified only for three subsequence study waves (covering nine months in total), the data do not support an event history analysis which is why we estimate the probability of having moved from employment to non-employment with logistic regression models. Our study also does not include links between developing Long COVID with unemployment or the receipt of social benefits. These are important areas for further research in order to capture the full impacts of the pandemic, and to better understand how best to support people with Long COVID in the workplace.

Several policy implications arise from this research. In the specific context of the UK, the extension of Statutory Sickness Pay beyond 28 weeks and greater flexibility to manage partial

(returns to) work would help to stem the flow out of employment of those with Long COVID. Financial support for employers to maintain employment until recovery from Long COVID would also help preserve employment.

Our findings further demonstrate that Long COVID poses a twin challenge to employers of providing adaptations and flexibility to manage the physical symptoms, as well as supporting improvements to mental health and well-being. There is also a legal point for employers in that a sizeable number of workers with Long COVID are likely to meet the definition of disability once they have had Long COVID for 12 months (following the Equality Act 2010 in the UK), and so have a right to receive reasonable workplace adjustments. Employers therefore have a lawful duty to consider and act on the needs of their workforce in relation to Long COVID once it reaches 12 months in duration.

Our research has demonstrated the importance of the workforce impacts of Long COVID. However, there is a need to better understand the diversity of employer responses and 'what works' in supporting employees with Long COVID. Organisational case studies of good Human Resource Management (HRM) practice around health management of fluctuating and comorbid health conditions, and examining inequalities in experiences, are needed to enable HRM practitioners to make the case for resources and focus on this area, as well to support compliance with legislation.

## Supporting information

**S1 Appendix. Sample description and additional models on work hours.**
(DOCX)

## Author Contributions

**Conceptualization:** Darja Reuschke, Donald Houston, Paul Sissons.

**Data curation:** Darja Reuschke.

**Formal analysis:** Darja Reuschke.

**Methodology:** Darja Reuschke, Donald Houston.

**Writing – original draft:** Darja Reuschke, Donald Houston, Paul Sissons.

**Writing – review & editing:** Darja Reuschke, Donald Houston, Paul Sissons.

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
