## [Decision Letter · Decision Letter 0]

22 Apr 2024

PONE-D-24-03100The impact of Long COVID on workers' employment and mental healthPLOS ONE

Dear Dr. Reuschke,

Thank you for submitting your manuscript to PLOS ONE. After careful consideration, we feel that it has merit but does not fully meet PLOS ONE’s publication criteria as it currently stands. Therefore, we invite you to submit a revised version of the manuscript that addresses the points raised during the review process.

We look forward to receiving your revised manuscript.

Kind regards,

Gabriel A. Picone

Academic Editor

PLOS ONE

Journal Requirements:

Reviewers' comments:

Reviewer's Responses to Questions

**Comments to the Author**

1. Is the manuscript technically sound, and do the data support the conclusions?

Reviewer #1: Yes

Reviewer #2: Yes

2. Has the statistical analysis been performed appropriately and rigorously? 

Reviewer #1: Yes

Reviewer #2: Yes

3. Have the authors made all data underlying the findings in their manuscript fully available?

Reviewer #1: Yes

Reviewer #2: Yes

4. Is the manuscript presented in an intelligible fashion and written in standard English?

Reviewer #1: Yes

Reviewer #2: Yes

5. Review Comments to the Author

**Reviewer #1:** Summary:

This paper investigates the impact of Long COVID-19 on employment and mental health using the Understanding Society COVID-19 Study 2020-2021 in the UK. Employing fixed and random effect models, the authors discovered that Long COVID does not influence employment status but significantly increases the likelihood of working zero hours. Additionally, Long COVID is associated with mental health issues and reduced happiness. The findings contribute to understanding broader health issues like chronic fatigue syndrome and offer policy implications for COVID-19 and similar health conditions on employment.

Major comments:

1. The context and existing evidence section extensively reviews the current literature, facilitating a comprehensive understanding of the topic. However, the authors did not explain why utilizing Long COVID would contribute to the existing literature. Consider adding more details and explaining the novelty/difference of employing Long COVID as opposed to using other health issues like depression and chronic pain.

2. Authors tried to differentiate the effect of the different durations of Long-term COVID-19 on various outcomes, and those different exposures would help us better understand the effect of Long-term COVID. Table 1 shows the total effect first, then the differential effect with different durations of employment. However, the authors did not do so in Table 2 and onward. Consider showing the total effect first (Long COVID compared to short COVID/No symptoms), then show the differential effect. Or clarify why the exposure/variables of interests are different for different outcomes.

3. For Table 1 and onward, the unique individuals of the fixed effect model are 407, and the random effect are 7092. Why those two numbers are so different? Does it mean that only 407 individuals repeated the survey, thus compared within person are much less? If so, can you please report the survey follow-up rate? If less than 10% of the survey respondents repeated the survey, would you consider selection bias might be a major weakness of this study?

4. In the conclusion section, authors acknowledge insufficient data for exploring Long COVID's impact on long-term employment. Consequently, the study concentrates solely on short-term effects. Consider aligning this limitation in the introduction/context sections and changing the narrative to short-term effects rather than giving an impression the study would study both.

5. The main paper flows well, but the abstract needs improvement for better clarity. Consider reworking it to include reporting insignificant results, clarify the comparison group, provide more details on the regression model, and succinctly highlight the study's contribution and policy implications.

Minor comments:

1. Suggest adding more details to the title: The Impact of Long COVID on Workers’ Employment and Mental Health: Empirical Evidence from the UK

2. Abstract line 33. “The results show that Group 2 is at higher risk of exiting employment.” What is the comparison group? Compared to those who did not have long COVID?

3. For section 3.1.1, can you add more details as to why you created four cohorts for this study? Do you expect/hypothesize people who experience long duration Long COVID, such as symptoms of 29 weeks or more, have a more significant impact on their employment and mental health? Why not study total effect: Long COVID or not despite the duration.

4. For section 3.1.1, can you confirm if someone had 4 weeks and 3 days of COVID symptoms, they will not be included? The current definition only includes 0 to 4 weeks, 5 to 28 weeks, or 29 weeks. Or is it included in the 5 to 28 weeks cohorts? Please make it clearer.

5. For section 3.1.2, do you define employment as those employed at the time of the survey? For most labor studies, we would define employment as either employed or looking for jobs at the time of the survey. Can you please clarify and explain why you defined employment in this way?

6. For section 3.2, can you add more details about why survey responses in waves 7-9 (January-September 2021) were chosen, not other waves?

**Reviewer #2:** The article makes relevant contributions related to an emerging problem in global public health, which is the burden of Covid-19 sequelae and its influence on occupational health.

There is a need for adjustments to improve the quality of the manuscript.

The methodology, with regard to evaluations and statistics, needs to be described more clearly to the reader. Details about which statistical program was carried out are missing.

Present the description of the tables more clearly in the manuscript. Highlight the values of significant findings in the text.

The discussion needs to be in-depth. What are the limitations of the study? It wasn't clear.

6. PLOS authors have the option to publish the peer review history of their article (what does this mean?). If published, this will include your full peer review and any attached files.

Reviewer #1: **Yes: **Lei Lv

Reviewer #2: No

---

## [Author Response · Author response to Decision Letter 0]

16 May 2024

REVIEWER #1

Major comments:

1. The context and existing evidence section extensively reviews the current literature, facilitating a comprehensive understanding of the topic. However, the authors did not explain why utilizing Long COVID would contribute to the existing literature. Consider adding more details and explaining the novelty/difference of employing Long COVID as opposed to using other health issues like depression and chronic pain.

Response: Long COVID is a new ‘disease’ or syndrome, often with relatively serious symptoms and substantial prevalence in the population, making it worthy of investigation in an empirical sense to establish an evidence baseline of its social patterns in prevalence, duration and impacts. Conceptually, investigating Long COVID contributes to existing understandings of complex comorbidity syndromes, which are often difficult or impossible to diagnose in a strict medical sense so remain under-recognised, under-reported and under-treated.

In the introduction, we reference figures on the scale of Long COVID and the increased risk of the working-age population to develop Long COVID. We add in section 2 (context and existing evidence), after reviewing relevant related literature on disability and employment on page 6 (lines 122-127), that Long COVID adds to previous studies on work-limiting conditions through focussing on an illness which is often associated, similar to chronic fatigue syndrome, with comorbid health conditions and people often do not receive appropriate treatment and are not eligible for disability benefits. The latter informs our novel measure of the duration of COVID-19 symptoms which is linked with the reviewer’s next comment.

2. Authors tried to differentiate the effect of the different durations of Long-term COVID-19 on various outcomes, and those different exposures would help us better understand the effect of Long-term COVID. Table 1 shows the total effect first, then the differential effect with different durations of employment. However, the authors did not do so in Table 2 and onward. Consider showing the total effect first (Long COVID compared to short COVID/No symptoms), then show the differential effect. Or clarify why the exposure/variables of interests are different for different outcomes.

Response: We have addressed this issue in various places to make clearer why we use a measure of Long COVID with different durations of symptoms.

We add this paragraph at the end of the context and literature section (section 2) on page 9 (lines 178-184):

“Together, these gaps in the sick pay system and inequalities in the coverage of occupational health support represent important limitations in the provision of support to allow individuals to remain in work while managing health conditions in the UK. However, no existing study has investigated the association between work outcomes of Long COVID and the length of employment protection in law and how the employment and in-work experiences of Long COVID may be mitigated by certain worker or job characteristics (e.g. skills level, earnings).”

We clarify further in section 3.1.1 on the use of measures (page 11, lines 225ff):

“Our focus is on the health-related COVID-19 impact on employment and in-work experiences of those who remain in work. Longer-term periods of ill health that extend the maximum period of Statutory Sickness Pay of 28 weeks in the UK, often precipitate the termination of employment [35]. This is the maximum period we expect especially low-paid workers and those in more insecure employment positions to be more likely to leave employment. Those on lower income and in lower skilled jobs in particular, may need to work even if they are unwell which may result in lower in-work experiences. This is the rationale for deriving two groups of individuals with Long COVID for this study: 1) 5-28 weeks of symptoms and 2) symptoms of 29 weeks and more … 

This is our preferred measure of Long COVID for this study. However, when estimating employment exit we also use a binary (total) measure of Long COVID (see section 3.3.1). This measure defines as Long COVID all those who reported that they have not recovered from a COVID-19 infection or who had experienced coronavirus symptoms for five weeks and more [32]. The comparison group on this binary measure are respondents with no COVID-19 symptoms or who recovered from a coronavirus infection.”

We completely revised section 3.3.1 (employment exit) as follows (page 16):

“We estimate the probability of having moved from employment to non-employment with logistic regression models. Unfortunately, we cannot fit a logistic regression model with fixed effects using our duration-based measure. The model requires that people have experienced between subsequent waves changes in the work status and COVID-19 symptoms categories. We can fit a logistic regression model with fixed effects using the binary (total) Long COVID measure (COVID-19 symptoms 5+ weeks vs <5 weeks including no symptoms and recovery from COVID-19). We report random-effects findings for the duration-based measure of Long COVID (no symptom, short COVID, Long COVID 5-28 weeks, Long COVID 29+ weeks).”

We delete in Table 2 (previously Table 1) the column showing estimates of the random-effects model with the binary Long COVID measure to be more consistent. We therefore consistently display random-effects findings only for the duration-based measure.

3. For Table 1 and onward, the unique individuals of the fixed effect model are 407, and the random effect are 7092. Why those two numbers are so different? Does it mean that only 407 individuals repeated the survey, thus compared within person are much less? If so, can you please report the survey follow-up rate? If less than 10% of the survey respondents repeated the survey, would you consider selection bias might be a major weakness of this study?

Response: These fixed effects models measure change. Model 1 in Table 1 (this is not Table 2) therefore only includes observations from individuals who exited employment and developed Long COVID or recovered from Long COVID. All individuals who are still in employment and who never developed Long COVID are excluded. Individuals who reported Long COVID in all waves (waves 7-9) are also excluded. This explains why the number of individuals is small compared to the random effects model that includes all observations. The advance of the fixed effects method is that it is quasi-experimental. The disadvantage is that those with continuous Long COVID are not included. We therefore also show findings from the random effects models where the latter are included.

We fully revise the section (3.3) that explains the analytical framework as follows (on pages 15-16):

“The fixed-effects panel model estimates within-individual variation and requires that the same individual is observed twice, once when they experienced COVID-19 symptoms and once when they did not experience COVID-19 symptoms. These models include an error term that represents individual-specific unmeasured characteristics and therefore allow to control for unobserved heterogeneity due to time-invariant omitted variables and traits. The disadvantage is, however, that individuals in our study who experienced Long COVID in all three subsequent waves, are not included in the statistical analysis.

This is why we also report random-effects models which estimate between-individual variation and within-individual variation. These models, however, assume that the error term is not correlated with unobserved time-invariant (and time-variant) variables. Random-effects models allow the investigation of time-invariant characteristics which we also report. We include as time-invariant variables pre-COVID-19 job characteristics of the respondents which we link to the survey data from the main panel study.”

We also expand the description of the data and include information on the retainment of participants in the sample (in sections 3.1 and 3.1.1) underlining the high-quality of the data. The retainment over the whole study comprising nine waves (18 months) is 40.5%. Across waves 7-9 which our analysis draws on, 63.5% remained in the study in each of the three waves (over nine months).

New text is added on pages 9-10 on the panel nature of the data (lines 198-203):

“In the first wave, 17,745 individuals completed the survey [41]. Individuals could also join the study in waves 2-4. In total, across all nine waves, 19,763 individuals participated in the study. However, as in all longitudinal surveys, there is an issue with panel attrition. From all individuals who participated in the study, 40.5% participated in every wave. A relatively high proportion of 9.5% only participated in wave 1 (April 2020).”

This information is added on page 10 specifically for the retention in waves 7-9 (lines 220-223):

“From the initially recruited participants, n= 15,249 were still in the study at least once over the waves 7-9 and n=9,678 individuals were in the study in each of these three waves our study is based on.”

4. In the conclusion section, authors acknowledge insufficient data for exploring Long COVID's impact on long-term employment. Consequently, the study concentrates solely on short-term effects. Consider aligning this limitation in the introduction/context sections and changing the narrative to short-term effects rather than giving an impression the study would study both.

Response: We agree with the reviewer. Consequently, we have changed the sentence in the introduction as follows (pages 5-6, lines 108-110)):

“We use a large national longitudinal survey of people 16 years and older and multi-variate panel regression modelling to estimate how the development of Long COVID has changed individuals’ employment and in-work experience.”

The short-term nature of the study is added in the conclusion in the context of discussing limitations of the study (on page 31, lines 626-631):

“Given the study design, we were only able to investigate short-term effects of developing Long COVID on employment and mental health but not the longer-term effects of Long COVID on people’s likelihood to return to, or remain in, work. Since Long COVID could be identified only for three subsequence study waves (covering nine months in total), the data do not support an event history analysis which is why we estimate the probability of having moved from employment to non-employment with logistic regression models.”

5. The main paper flows well, but the abstract needs improvement for better clarity. Consider reworking it to include reporting insignificant results, clarify the comparison group, provide more details on the regression model, and succinctly highlight the study's contribution and policy implications.

Response: Many thanks for these suggestions. In response, we now use a structured abstract approach in order to succinctly cover all these points. The reference group is added.

Minor comments:

1. Suggest adding more details to the title.

Response: The title has been amended to provide more detail:

“Impacts of Long COVID on workers: a longitudinal study of employment exit, work hours and mental health in the UK”

2. Abstract line 33. “The results show that Group 2 is at higher risk of exiting employment.” What is the comparison group? Compared to those who did not have long COVID?

Response: The comparison group is now added.

3. For section 3.1.1, can you add more details as to why you created four cohorts for this study? Do you expect/hypothesize people who experience long duration Long COVID, such as symptoms of 29 weeks or more, have a more significant impact on their employment and mental health? Why not study total effect: Long COVID or not despite the duration.

Response: We can see that the reference to the 28 weeks of Statutory Sickness Pay in Section 3.1.1 was too brief and that the use of an acronym was not helpful (“Given our focus on the health-related COVID-19 impact on employment, we further identify whether respondents have suffered from Long COVID for more than 28 weeks corresponding to the maximum period of SSP.”). We therefore changed the text on page 11 (lines 225ff.) as follows:

“Our focus is on the health-related COVID-19 impact on employment and in-work experiences of those who remain in work. Longer-term periods of ill health that extend the maximum period of Statutory Sickness Pay of 28 weeks in the UK, often precipitate the termination of employment [35]. This is the maximum period we expect especially low-paid workers and those in more insecure employment positions to be more likely to leave employment. Those on lower income and in lower skilled jobs in particular, may need to work even if they are unwell which may result in lower in-work experiences. This is the rationale for deriving two groups of individuals with Long COVID for this study: 1) 5-28 weeks of symptoms and 2) symptoms of 29 weeks and more. For comparison, we define a further two groups of respondents with 3) COVID-19 symptoms of up to four weeks (short COVID) after a COVID-19 infection and 4) a group with no symptoms including recovery from a previous coronavirus infection (used as reference group in the modelling). This is our preferred measure of Long COVID for this study. However, when estimating employment exit we also use a binary (total) measure of Long COVID (see section 3.3.1). This measure defines as Long COVID all those who reported that they have not recovered from a COVID-19 infection or who had experienced coronavirus symptoms for five weeks and more [32]. The comparison group on this binary measure are respondents with no COVID-19 symptoms or who recovered from a coronavirus infection.”

4. For section 3.1.1, can you confirm if someone had 4 weeks and 3 days of COVID symptoms, they will not be included? The current definition only includes 0 to 4 weeks, 5 to 28 weeks, or 29 weeks. Or is it included in the 5 to 28 weeks cohorts? Please make it clearer.

Response: The question about the length of COVID-19 symptoms was: “For how many weeks have you experienced coronavirus symptoms?” with the response: [numeric textbox] weeks (see page 9). There were no instructions added to this question. Values ranged from 1-60 and therefore respondents could not answer 4.5 weeks, for example, if they had symptoms for 4 weeks and 3 days. We can therefore only assume that most people will have provided answers for the full week, i.e. in the case of 4 weeks and 3 days they would have chosen 4 weeks. We also changed the wording and now refer in the text to having coronavirus symptoms for five weeks and more instead of saying more than 4 weeks – which we agree may be confusing.

We are grateful for this comment as this points to a potential minor bias of the study addressing also the other reviewer’s feedback on limitations of the data. We add this issue as a limitation of the data in the discussion section (page 31, 622-624).

5. For section 3.1.2, do you define employment as those employed at the time of the survey? For most labor studies, we would define employment as either employed or looking for jobs at the time of the survey. Can you please clarify and explain why you defined employment in this way?

Response: Employment is defined in this study as being in employment (paid employment or self-employment). The unemployed are not included in this definition. This is conceptually meaningful in our study as we investigate the association between Long COVID and the risk of leaving employment, but statistically the unemployed are not in the population at risk of this outcome (i.e. one needs to be employed to be at risk of leaving employment).

The data would also not allow us to differentiate between those who are economically active (employed or unemployed) or economically inactive as the possible responses to the question “Thinking about your situation now. Even if you did not do any paid work last week, are you currently employed or self-employed?” are as follows: 1) Yes, employed only, 2) yes, self-employed only, 3) both employed and self-employed, 4) No. Both the unemployed and economically inactive fall into the fourth category.

We fully revise the first paragraph in section 3.1.2 (measuring em

---

## [Decision Letter · Decision Letter 1]

12 Jun 2024

Impacts of Long COVID on workers: a longitudinal study of employment exit, work hours and mental health in the UK

PONE-D-24-03100R1

Dear Dr. Reuschke,

We’re pleased to inform you that your manuscript has been judged scientifically suitable for publication and will be formally accepted for publication once it meets all outstanding technical requirements.

Kind regards,

Gabriel A. Picone

Academic Editor

PLOS ONE

Additional Editor Comments (optional):

Reviewers' comments:

Reviewer's Responses to Questions

**Comments to the Author**

1. If the authors have adequately addressed your comments raised in a previous round of review and you feel that this manuscript is now acceptable for publication, you may indicate that here to bypass the “Comments to the Author” section, enter your conflict of interest statement in the “Confidential to Editor” section, and submit your "Accept" recommendation.

Reviewer #1: All comments have been addressed

2. Is the manuscript technically sound, and do the data support the conclusions?

Reviewer #1: Yes

3. Has the statistical analysis been performed appropriately and rigorously? 

Reviewer #1: Yes

4. Have the authors made all data underlying the findings in their manuscript fully available?

Reviewer #1: Yes

5. Is the manuscript presented in an intelligible fashion and written in standard English?

Reviewer #1: Yes

6. Review Comments to the Author

Reviewer #1: (No Response)

7. PLOS authors have the option to publish the peer review history of their article (what does this mean?). If published, this will include your full peer review and any attached files.

Reviewer #1: No

---

## [Editor Report · Acceptance letter]

17 Jun 2024

PONE-D-24-03100R1 

PLOS ONE

Dear Dr. Reuschke, 

I'm pleased to inform you that your manuscript has been deemed suitable for publication in PLOS ONE. Congratulations! Your manuscript is now being handed over to our production team.

Kind regards, 

on behalf of

Dr. Gabriel A. Picone 

Academic Editor

PLOS ONE